# The Transcription Factor *ZmNAC89* Gene Is Involved in Salt Tolerance in Maize (*Zea mays* L.)

**DOI:** 10.3390/ijms242015099

**Published:** 2023-10-12

**Authors:** Yingying Hu, Chunxiang Li, Runyu Zhou, Yongfeng Song, Zhichao Lv, Qi Wang, Xiaojie Dong, Shan Liu, Chenchen Feng, Yu Zhou, Xing Zeng, Lin Zhang, Zhenhua Wang, Hong Di

**Affiliations:** 1Key Laboratory of Germplasm Enhancement, Physiology and Ecology of Food Crops in Cold Region, Engineering Technology Research Center of Maize Germplasm Resources Innovation on Cold land of Heilongjiang Province, Northeast Agricultural University, Harbin 150030, China; com12569@163.com (Y.H.); lcx990127@163.com (C.L.); zhourunyu19991005@163.com (R.Z.); song18182746326@163.com (Y.S.); lvzhichao2004@163.com (Z.L.); wwangqi097@163.com (Q.W.); dongxiaojie0605@163.com (X.D.); m17653057335@163.com (S.L.); 18324508201@163.com (C.F.); zhouyu0924@126.com (Y.Z.); zengxing980@hotmail.com (X.Z.); neauzla@163.com (L.Z.); 2Institute of Crop Resources Research, Heilongjiang Academy of Agricultural Sciences, Harbin 150086, China

**Keywords:** maize, saline–alkaline tolerance, *ZmNAC89* gene, RNA-sequencing analysis

## Abstract

The NAC gene family has transcription factors specific to plants, which are involved in development and stress response and adaptation. In this study, *ZmNAC89*, an NAC gene in maize that plays a role in saline–alkaline tolerance, was isolated and characterized. ZmNAC89 was localized in the nucleus and had transcriptional activation activity during in vitro experiments. The expression of *ZmNAC89* was strongly upregulated under saline–alkaline, drought and ABA treatments. Overexpression of the *ZmNAC89* gene in transgenic *Arabidopsis* and maize enhanced salt tolerance at the seedling stage. Differentially expressed genes (DEGs) were then confirmed via RNA-sequencing analysis with the transgenic maize line. GO analyses showed that oxidation–reduction process-regulated genes were involved in *ZmNAC89*-mediated salt–alkaline stress. *ZmNAC89* may regulate maize saline–alkali tolerance through the REDOX pathway and ABA signal transduction pathway. From 140 inbred maize lines, 20 haplotypes and 16 SNPs were found in the coding region of the *ZmNAC89* gene, including the excellent haplotype HAP20. These results contribute to a better understanding of the response mechanism of maize to salt–alkali stress and marker-assisted selection during maize breeding.

## 1. Introduction

One of the most important abiotic stressors is soil salinization, which hinders crop yields and can destabilize the global food supply [1]. Maize is an important food crop, accounting for more than half of the calories consumed globally [2]. However, maize is a glycophyte and is hypersensitive to salt stress [3]. As such, it is important to understand how maize tolerates salt and to identify new candidate genes in maize breeding.

The NAC transcription factor is an acronym of the *NAM*, *ATAF1/2* and *CUC2* genes. The first *NAC* gene, *NAM* (*No Apical Meristem*), is from *Petunia hybrida* and is related to the development of the shoot tip meristem [4]. Subsequently, *ATAF1/2* and *CUC2* genes similar to *NAM* structures were found in *Arabidopsis thaliana* [5]. Bioinformatics analyses indicated that the NAC transcription factor has 151 members in rice (*Oryza sativa*) [6], 117 members in *Arabidopsis* [7] and 152 members in tobacco (*Nicotiana tabacum*) [8] and soybean (*Glycine max*) [9]. Analysis of expression profiling demonstrated that 20–25% of NAC genes were activated by a minimum of one abiotic stressor, such as drought conditions, freezing temperatures, salt or phytohormones and ABA [10]. These helped regulate response gene expression, which contributes to abiotic stress resistance in plants.

Combined with the promoter of the ABA signal regulator *OsbZIP23*, *SNAC1* positively regulated the expression of downstream ABA signal genes to improve drought and salt tolerance and increase rice yield [11,12]. Its homologous gene *SNAC2* can improve salt and cold resistance [13]. *SNAC3* improves drought and heat resistance by regulating ROS detoxification genes in rice [14] *OsNAC5*, *6* and *10* improved drought tolerance and reduced yield loss [15,16,17]. *OsNAC66* can improve rice’s drought resistance and antioxidant ability and resist rice blast and leaf blight by inhibiting the ABA signal pathway [18,19]. *OsNAC2* depends on the ABA pathway to accelerate leaf senescence in rice [20]. Other reports indicate that NAC genes can improve crop resistance, including chickpea *CarNAC2* [21], sorghum *SbSNAC1* [22], wheat *TaNAC29* [23,24] and *TaNAC2-5A* [25], tomato *SlNAP2* [26] and *SlJUB1* [27]. This indicates that NAC transcription factors have potential applications in abiotic stress breeding.

Based on the sequence characteristics of NAC protein, Voitsik et al. found 116 NAC genes by scanning the whole genome of maize and cloned two NAC transcription factor genes, *ZmNAC41* and *ZmNAC100*, which are related to maize anthracnose [28]. Previously, the functions of some maize NAC transcription factor genes have been verified: *ZmSNAC1*, *ZmNAC33* and *ZmNAC55* significantly increase the drought tolerance of transgenic *Arabidopsis* [29,30,31]. A genome-wide association study confirmed that the *ZmNAC111* gene is significantly associated with drought resistance. The promoter region has an insertion of a miniature inverted repeating transposon MITE [32]. *ZmaNAC34* and *ZmNAC36* genes are associated with starch synthesis [33,34]. *ZmNAC84* was phosphorylated by *ZmCCaMK* at Ser-113 and induced antioxidant defense by activating downstream genes [35].

Previously, we analyzed differentially expressed genes in the maize transcriptome under alkaline–salt Na_2_CO_3_ stress using second-generation high-throughput sequencing RNA-Seq technology [36]. We found that the *ZmNAC89* gene significantly increased expression levels under stress conditions. In this study, bioinformatics analysis, gene expression pattern analysis under stress treatment, genetic transformation and RNA-seq were used to achieve the following goals: (1) identify the basic characteristics of *ZmNAC89* gene, (2) identify the expression of *ZmNAC89* gene in other abiotic stresses, (3) identify the best haplotype, as well as predict and identify the extremely saline–alkali-tolerant maize inbred lines for future breeding applications and (4) provide an initial understanding of the response mechanism of maize to saline–alkali stress in the breeding process.

## 2. Results

### 2.1. Phylogenetic Analysis of the ZmNAC89 Gene

The *ZmNAC89* gene (Gene ID: *GRMZM2G430849*, XM_008653325.1) is localized in the bin 7.05 region based on B73 RefGen_v2 genome-wide data. Sequence analysis indicated that full-length *ZmNAC89* is 2280 bp and that it possesses a common PolyA tail signal (AATAAA) +158 bp~+164 bp after the stop codon TAG (Appendix A). We identified 31 orthologs of ZmNAC89 in different species from *Arabidopsis*, rice, wheat and soybean and constructed the phylogenetic tree of ZmNAC89 orthologs. The ZmNAC89 protein is 76% similar to the AtNAP protein associated with salt stress in *Arabidopsis thaliana*. It is in the same clade as the OsNAC10 protein related to salt stress in rice and the TaNAC3 protein related to drought resistance in wheat (Figure 1 and Appendix A). Meanwhile, the *ZmNAC89* promoter has core elements, such as CAAT-box and TATA-box, as well as biotic and abiotic stress-related elements (Appendix A). Therefore, the promoter could be a stress-inducible promoter.

### 2.2. Transactivation Activity Analysis of ZmNAC89

In order to verify whether *ZmNAC89* has transcriptional activation activity, bait plasmid pGBKT7-*ZmNAC89*/pGADT7-T, positive control plasmid pGBKT7-53/pGADT7-T and negative control plasmid pGBKT7-Lam/pGADT7-T were, respectively, co-transformed into Y2HGlod yeast cells, and the growth status and color change in colonies were observed on the selection medium to judge whether the bait plasmid has transcriptional activity. Blue spot colonies appeared in bait plasmid pGBKT7-*ZmNAC89*/pGADT7-T and positive control plasmid pGBKT7-53/pGADT7-T, indicating that the *ZmNAC89* transcription factor has transcriptional activation activity (Figure 2).

### 2.3. Subcellular Localization of the ZmNAC89 Protein

To identify the subcellular localization of the ZmNAC89 protein, we transformed the recombinant plasmid into the protoplast of Arabidopsis thaliana and observed the fluorescence signal by using laser confocal microscopy. The green fluorescence signal of the control sample transformed into the empty vector was present in the nucleus and the cytoplasmic region of the protoplast. However, the green fluorescence signal of the ZmNAC89-GFP fusion protein was located in the nucleus of the protoplast. This suggested that the ZmNAC89 protein is localized on the nucleus (Figure 3).

### 2.4. Inducible Expression of ZmNAC89 in Response to Various Abiotic Stresses

To examine the function of *ZmNAC89*, we analyzed the expression pattern of the *ZmNAC89* gene in ‘K10′ inbred lines under salt, alkalinity, ABA and drought stress conditions. The *ZmNAC89* transcript level was induced through treatment with NaCl, Na_2_CO_3_, ABA and PEG in the roots and leaves. Under NaCl and Na_2_CO_3_ treatments, the expression of *ZmNAC89* achieved a maximum level at 3 and 24 h in the leaves and roots, respectively (Figure 4A,B). Under ABA treatment, the transcript level of *ZmNAC89* reached a peak at 3 and 12 h in the leaves and roots, respectively (Figure 4C). Under PEG treatment, peaks appeared at 6 h and 9 h in the leaves and roots, respectively (Figure 4D). These results indicated that *ZmNAC89* responds to various abiotic stresses and could have a prominent function in salt and alkalinity adversities.

### 2.5. ZmNAC89 Gene Overexpression Increased Arabidopsis and Maize Resistance to Alkalinity and Salt

*ZmNAC89* expression levels are related to maize salt–alkaline tolerance. Therefore, we produced transgenic maize and *Arabidopsis*, both of which overexpressed the *ZmNAC89* coding sequence (from the K10 genotype). The *ZmNAC89* cDNA clone sequence was sent to GenBank (Accession No. MK125509.1).

The phenotypes of *ZmNAC89* lines during germination were analyzed for transgenic *Arabidopsis*. Under 100 mM NaCl stress, the root length, root surface area and root volume of the transgenic lines were significantly greater than the wild type (6.31 cm, 0.29 cm^2^ and 0.00064 cm^3^, respectively), with an average of 9.74 cm, 0.4 cm^2^ and 0.00099 cm^3^ (Figure 5A,C,G). After 150 mM NaCl treatment, the root length and root surface area were significantly greater in the transgenic lines than in the wild type (2.24 cm and 0.09 cm^2^, respectively), with an average of 3.33 cm and 0.12 cm^2^ (Figure 5A,C). Under 0.5 mM NaHCO_3_ stress treatment, the root length, root surface area and root volume of the transgenic lines were significantly higher than in the wild type (7.72 cm, 0.29 cm^2^ and 0.00071 cm^3^, respectively), with an average of 9.38 cm, 0.39 cm^2^ and 0.00095 cm^3^, respectively (Figure 5B,D,H). After 1 mM NaHCO_3_ stress treatment, the root mean diameter was significantly higher in the transgenic lines than in the wild type (0.098 mm), with an average of 0.12 mm (Figure 5F). To further test the function of *ZmNAC89* in salt and alkalinity tolerance, we observed the phenotypes of the transgenic *Arabidopsis* in the mature stage. After salt and alkali treatments, the transgenic lines still maintained solid growth and produced yellow-green leaves, erect stems and normal flowering and fruit. However, the wild type showed wilting, white leaves, curved stems and fewer flowers (Figure 6).

The stress resistance of *ZmNAC89* transgenic lines could be reflected by measuring physiological indexes related to stress, such as relative conductivity, chlorophyll and proline. Under 300 mM NaCl stress or 60 mM NaHCO_3_ stress, the six transgenic lines had a relative conductivity that was significantly lower than wild-type *Arabidopsis*. In contrast, the proline and chlorophyll contents were significantly higher in all six transgenic lines than in the wild type (Table 1 and Table 2). These results suggested that overexpression of the *ZmNAC89* gene can significantly improve the salt and alkalinity tolerance of *Arabidopsis thaliana*.

Improvements in alkalinity and salt tolerance were also observed in transgenic maize when the *ZmNAC89* gene was overexpressed. The germination-stage maize was treated with salt, alkali and saline–alkali mixed stresses, after which multiple comparisons of germination potential, germination rate, root length, shoot length, fresh weight and dry weight of each line were performed through Duncan analysis using SPSS20.0 software. Under different stress treatments, the index characteristics of each transgenic line showed that the germination potential, germination rate, root length, shoot length, fresh weight and dry weight of the transgenic lines DNAC89-C-3, DNAC89-C-5 and DNAC89-C-25 performed better than other transgenic lines and were significantly higher than the control line C01 (Appendix A–C). The seedling-stage maize was treated with salt, alkali and saline–alkali mixed stresses. Multiple comparisons of plant height, dry weight of the shoots and root dry weight were performed with Duncan analysis using SPSS20.0 software. The significance of differences between transgenic lines and C01 were analyzed using the LSD method. Under different stress treatments, the transgenic lines DNAC89-C-3, DNAC89-C-5 and DNAC89-C-25 were the most prominent in plant height, dry weight in the ground, root dry weight, total root length, root surface area and root volume (Figure 7A–D). The measurement results of relative conductivity, chlorophyll and proline content demonstrated that transgenic lines improved salt–alkali resistance compared to the controls. The relative conductivity of 17 overexpressing lines decreased while chlorophyll and proline content increased. DNAC89-C-3, DNAC89-C-5 and DNAC89-C-25 had better salt–alkali resistance than the other transgenic lines (Appendix A–C).

### 2.6. RNA-Seq Analysis and Prediction of Interacting Proteins

To better understand the molecular mechanism of the *ZmNAC89* gene responsible for regulating salt–alkali resistance, we compared transcriptomic data from the WT and transgenic *ZmNAC89* maize lines under Na_2_CO_3_ and NaCl conditions. edgeR software (3.42.4) was used to analyze the DEGs (|log2foldchange| ≥ 1, *p* < 0.05). Compared with the WT, the DEGs of transgenic lines increased significantly after stress treatment, and there were more downregulated DEGs than the upregulated DEGs, indicating that the expression of these genes was strongly regulated by *ZmNAC89* (Appendix A–D). Under NaCl treatment, 114 genes were upregulated and 140 were downregulated in the WT, while 932 genes were upregulated and 3137 were downregulated in the transgenic line (Figure 8A). Under Na_2_CO_3_ treatment, 326 genes were upregulated and 708 were downregulated in the WT, while there were 1008 upregulated and 5336 downregulated genes in the transgenic control line (Figure 8B). Wayne diagram analysis demonstrated that under different treatments of NaCl and Na_2_CO_3_, the transgenic lines had significantly differentially expressed genes compared with the receptor control lines. Under NaCl treatment, 3940 genes were only expressed in the transgenic lines (Figure 8C). Under Na_2_CO_3_ treatment, 5816 genes were only expressed in transgenic lines (Figure 8D). These results will help facilitate future analysis of the mechanism regulating *ZmNAC89* saline–alkali resistance.

To study the salt–alkaline tolerance network of maize regulated by *ZmNAC89*, a GO analysis was performed for significantly upregulated and downregulated genes. Under NaCl treatment, many DEGs involved in pathways for the L-phenylalanine catabolic process, oxidation–reduction process, transferase activity and chloroplast were highly enriched (Appendix A). Under Na_2_CO_3_ treatment, many DEGs involved in pathways for photosynthesis, light reaction, chloroplasts, translation and the oxidation–reduction process were highly enriched (Appendix A). GO analyses showed that oxidation–reduction process-regulated genes were involved in *ZmNAC89*-mediated salt–alkaline stress, and the *ZmNAC89* transcript level was induced by ABA. It has been reported that many MYB [37,38,39] and bZIP [40,41] transcription factors regulate the response to salt–alkaline tolerance through the oxidation–reduction process or ABA pathway. In combination with DEGs screened with RNA-seq, the MYB transcription factor (*Zm00001d014364*) and bZIP transcription factor (*Zm00001d044546*) were selected as candidate proteins and for further verification.

Under NaCl and Na_2_CO_3_ treatments, the expression of *Zm00001d014364* and *Zm00001d044546* (Figure 9) gradually increased in roots and leaves and peaked at 3 h and 24 h, respectively. Combined with the induced expression analysis of *ZmNAC89*, there was a positive correlation with *ZmNAC89* expression levels, indicating that there could be a positive regulatory relationship between them.

### 2.7. Sequence Variation Analysis of ZmNAC89

The sequence alignment of the *ZmNAC89* gene-coding region in 140 maize inbred lines showed that the sequence similarity of the *ZmNAC89* gene-coding region in different inbred lines was 99.87%, indicating that the coding region was relatively conservative. The polymorphic sites in this region were analyzed using 100 bp as the sliding frame and 25 bp as the step size. We found approximately 1–500 bp in the first exon region, indicating that there are abundant variations in this region and that polymorphism is significantly higher than in the other two exon regions. The maximum π value of nucleotide polymorphism in the second exon region reached 0.005, and there was no sequence variation in the third exon. This indicates that these two parts are relatively conservative (Figure 10).

A total of 16 SNP sites were detected in the coding region of the *ZmNAC89* gene of the tested maize inbred lines. The polymorphic sites were typed according to the mutation type and the number of nucleotides (Appendix A). The SNPs could be divided into six types: base C/G, G/A, G/T, T/C, A/G and C/T. DNA SPV6.0 software analysis was used to detect 20 haplotypes, with a polymorphism of 0.74. The main haplotypes were HAP1, HAP2 and HAP20, which collectively accounted for 82.14% of the tested materials. There were few other haplotypes, most of which belonged to rare variations. There were 40 Hap1 haplotype inbred lines, primarily including Han21, shen3336 and huotanghuang, and other saline–alkali-sensitive materials; the HAP2 haplotype contained 21 inbred lines, mainly guan17, 31 and Luyuan 133, and other moderately saline–alkali-tolerant materials; the HAP20 haplotype contained 54 inbred lines, mainly salt- and alkali-resistant materials such as Shen 118 and Dan 3130. The six SNP sites were synonymous mutations, which did not affect the amino acid sequence. The 10 SNP loci were non-synonymous mutations, which could change the structure and physiological function of proteins and affect resistance (Appendix A). According to the classification of salt- and alkali-resistant inbred lines, the HAP20 haplotype was preliminarily identified as an excellent haplotype.

Combined with the previous NaCl and NA_2_CO_3_ stress treatment, the identification of saline–alkali-tolerance-related traits at the seedling stage and comprehensive evaluation via the membership function method obtained clustering results, and the minimum allele frequency in DNAMAN comparison results was 5%. DNA SPV6.0 software was used for SNP nucleotide diversity analysis and haplotype diversity analysis. The results are shown in Table 3.

Under Na_2_CO_3_ stress, nucleotide diversity π = 0.00291 and Tajima’s D value was −1.10181 in 51 medium-resistance materials. Nucleotide diversity π = 0.00205 and Tajima’s D value was −0.06472, or nucleotide diversity π = 0.00286 and Tajima’s D value was −0.52477, in 23 resistant materials. Nucleotide diversity π = 0.00127 and Tajima’s D value was −0.53609 in 17 sensitive materials. Nucleotide diversity π = 0.00266 and Tajima’s D value was −0.15008 in 16 highly sensitive materials.

Under NaCl stress, nucleotide diversity π = 0.0024 and Tajima’s D value was 0.01116 in 45 resistant materials. Nucleotide diversity π = 0.00216 and Tajima’s D value was −1.03859 in 30 sensitive materials. Nucleotide diversity π = 0.00307 and Tajima’s D value was −0.82036 in 27 highly sensitive materials. Nucleotide diversity π = 0.00245 and Tajima’s D value was −0.38437 in 20 medium-resistance materials. Nucleotide diversity π = 0.00222 and Tajima’s D value was −0.67509 in 18 high-resistance materials.

Under Na_2_CO_3_ and NaCl stress, the nucleotide diversity π in maize inbred lines with different salt and alkali tolerance was relatively small, indicating that changes in the coding region of the *ZmNAC89* gene in maize inbred lines were relatively conservative. Tajima’s D value, Fu and Li’s D * value and Fu and Li’s F * value did not reach significant levels, indicating that the gene follows neutral evolution in the maize inbred line population.

## 3. Discussion

Maize salt–alkali tolerance is a complex trait, including different mechanisms of osmosis, ions, oxidation and others, so it is difficult to identify various genes that regulate it [35,42,43]. The NAC transcription factor is involved in abiotic stress responses in plants, including those to salinity, drought and low temperatures. Based on previous transcriptome data [36], we found that the *ZmNAC89* gene was significantly induced by alkaline–salt Na_2_CO_3_ stress. In this study, we identified *ZmNAC89* as a transcription factor with transcriptional activation activity and localization in the nucleus. The expression of *ZmNAC89* was strongly upregulated under saline–alkaline, drought and ABA treatments. This indicates that NAC transcription factor genes are involved in plant responses to abiotic stresses.

Overexpression of the *ZmNAC89* gene in transgenic *Arabidopsis* and maize enhanced salt tolerance at the germination and seedling stages. The rice *OsNAC5* and *OsNAC6* genes can improve salt and drought tolerance of rice by expanding root diameter and enhancing root development, and they regulate the expression of peroxidase genes and genes related to various drought tolerance pathways [15,16]. Under saline–alkali stress, the root length, average root diameter, root surface area and root volume of overexpressing lines were significantly higher than those of wild-type lines, indicating that *ZmNAC89* gene may improve salt–alkali tolerance by enhancing root development of plants.

Optimal haplotypes are typically mined based on gene sequence, genome sequence and phenotype screening. Haplotypes can be used to classify germplasm resources [44,45]. Based on the sequence variation analysis of the coding region of the *ZmNAC89* gene of 140 maize inbred lines, 20 haplotypes were identified, including 10 non-synonymous mutation sites. The results of the association analysis demonstrated that HAP20 was an excellent haplotype for salt and alkali tolerance. There is the potential to use the excellent saline–alkali-resistance inbred lines in HAP20 for breeding work, to use the information about excellent haplotypes to quickly identify the saline–alkali resistance of unknown resistant materials and to use this information to improve the speed of breeding applications. Maize inbred lines with this haplotype were identified, which is important when breeding salt and alkali tolerance.

Different cis-acting elements in the promoter region can regulate the expression of NAC transcription factors. The *OsNAC045* gene is related to drought resistance and salt tolerance in rice and is partially regulated by MYB and MYC elements [46]. *PbeNAC1* could increase the expression of some stress-related genes by interacting with *PbeDREB1* and *PbeDREB2A* to improve stress resistance in pears [47]. In this study, *Zm00001d014364* and *Zm00001d044546* genes as candidate interaction genes for ZmNAC89 protein were the MYB transcription factor and bZIP transcription factor, respectively. qRT-PCR was performed on *Zm00001d014364* and *Zm00001d044546* genes, which preliminarily demonstrated that their expression was positively correlated with *ZmNAC89*. The gene promoter region contained elements regulating abscisic acid, methyl jasmonate, gibberellin and auxin and the light response. However, further experiments are needed to determine the interaction protein of ZmNAC89.

Generally, abiotic stress is polytropic. *GmNF-YC14* can be significantly induced and upregulated under drought, salt stress and ABA treatment at the same time, and *GmNF-YC14* can form complexes with *GmNF-YB2* and *GmNF-YA16* in soybean plants [48]. In this study, the expression of *ZmNAC89* was upregulated under salt, alkali, drought and ABA stress, and the contents of proline and chlorophyll of transgenic lines under salt and alkali stress were increased. In addition, proline is a plant cytoplasmic osmotic regulator. It also plays an important role in stabilizing the structure of biological macromolecules, reducing cell acidity, detoxifying ammonia and regulating cell REDOX potential as an energy reservoir. The results in the transcriptome showed that differentially expressed genes were enriched into REDOX-related pathways and contained multiple ABA-signaling-related genes. Salt, alkali, drought and ABA can be generally classified as hypoxia and osmotic pressure imbalance-related stresses. We speculate that the *ZmNAC89* gene may regulate maize salt–alkali tolerance through the REDOX and ABA pathways, and that it may respond to drought and ABA stress as a transcription factor binding to downstream proline- or REDOX-related genes.

Abscisic acid, stress and maturation induction (ASR) genes are a class of plant-specific transcription factors, which play an important role in plant development, growth and abiotic stress response [49]. Whole-genome identification has been conducted in wheat and maize, and some genes in the family were found to be upregulated under stress treatment. *ZmNAC89* is also upregulated under ABA stress, which may act on the NAC family binding sites in *ZmASR*s promoters. The *ERECTA* gene is a candidate gene with a potential role in drought stress tolerance in soybean [50], and the *ERECTA* gene may also be activated by NAC family genes in maize. ZmNAC89 protein candidate interaction genes *Zm00001d014364* and *Zm00001d044546* are MYB transcription factors and bZIP transcription factors, respectively, and many MYB and bZIP transcription factors have been previously reported to improve the plant response to saline-based stress by regulating ABA pathways.

## 4. Materials and Methods

### 4.1. Plant Materials and Vector Plasmid

*A. thaliana* seeds (wild-type Col-0 (WT)) were provided by Pro. Shuzhen Zhang of the Soybean Research Institute of Northeastern Agricultural University. The maize inbred line C01 used as a transgenic receptor was provided by the Life Science and Technology Center of China Seed Group Co., Ltd. (Beijing, China). The binary vector pTF101 was provided by Pro. Kan Wang (Iowa State University, Ames, IA, USA).

### 4.2. Bioinformatics Analysis of ZmNAC89 Gene

The *ZmNAC89* gene was predicted with the online prediction software FGENESH (2023) (http://www.softberry.com/ accessed on 20 April 2023), using the genome-wide sequence of the maize inbred line B73 RefGen_v4 in the MaizeGDB database (http://www.maizegdb.org/ accessed on 20 April 2023). The neighbor-joining (NJ) method was used to build a phylogenetic tree of the aligned protein sequences in MEGA (X) (bootstrap value of 1000). The cis-acting elements of the *ZmNAC89* gene flanking sequence were deduced using PlantCARE (http://bioinformatics.psb.ugent.be/webtools/plantcare/html/ accessed on 20 April 2023).

### 4.3. Cloning of the Full-Length cDNA

Total RNA was extracted from seedlings of maize inbred line K10 treated with 150 mM NaCl for 3 h using TRIzol reagent (TransGen, Beijing, China). First-strand cDNA was synthesized from 1 μg of total RNA with 1 μL (200 U) TransScript^TM^ RT/RI Enzyme Mix (TransGen), according to the manufacturer’s instructions. The primers for *ZmNAC89* were performed using KOD-Plus-Neo (TOYOBO, Osaka, Japan) under the following conditions: 94 °C for 3 min, followed by 30 cycles of 94 °C for 45 s, 58.3 °C for 30 s, 72 °C for 1 min and 72 °C for 10 min. The resulting amplification product was subjected to gel purification and cloned to the pEASY-T1 Simple vector (TransGen), after which it was sequenced. The primers are listed in Appendix A.

### 4.4. Transactivation Assay

The *ZmNAC89* coding sequence was cloned into the pGBKT7 vector. The pGBKT7-ZmNAC89 constructs and the empty vector pGADT7-T were co-transformed into Y2HGlod yeast cells. The empty vector pGADT7-T and pGBKT7-53 were co-transformed as positive controls. The empty vector pGADT7-T and pGBKT7-Lam were co-transformed as negative controls. Transcriptional activation activity was determined on synthetically defined (SD)/-Ade/-His/-Leu/-Trp/X-ɑ-Gal/AbA medium.

### 4.5. Subcellular Location of ZmNAC89

The *ZmNAC89* coding sequence fused with GFP was cloned into the pCAMBIA1302 vector, and then pCAMBIA1302-ZmNAC89-GFP and pCAMBIA1302-GFP were transformed into *Arabidopsis* protoplasts. After 16 h incubation at 25 °C, the GFP fluorescent signal of the transfected *Arabidopsis* protoplasts cells was examined with a TCS SP2 confocal spectral microscope imaging system (Leica, Wetzlar, Germany).

### 4.6. Analysis of Inducible Expression Using Real-Time Quantitative PCR

For qRT–PCR, total RNA was extracted from third-leaf-stage K10 seedlings treated with 15 mM Na_2_CO_3_, 150 mM NaCl, 20% (*w*/*v*) PEG6000 and 100 μM ABA for 0 h, 1 h, 3 h, 6 h, 9 h, 12 h, 24 h, 2 d and 4 d using TRIzol reagent (TransGen). cDNA was synthesized using TransScriptTM RT/RI Enzyme Mix (TransGen), and qRT–PCR analyses were conducted using TransStart^®^ Tip Green qPCR SuperMix (TransGen) on a Chromo4^TM^ Real-Time Detector (Bio-Rad, Hercules, CA, USA). The maize *Actin1* gene (Gene ID: 100282267) was used as an internal standard. The 2^−ΔΔCt^ method was used to analyze relative expression. The primers are listed in Appendix A.

### 4.7. Construction and Transformation of Binary Plant Expression Vector

To generate transgenic lines overexpressing *ZmNAC89*, the coding sequence of *ZmNAC89* was cloned into the pTF101 vector and transformed into the *Agrobacterium tumefaciens* strain EHA105 to infect *Arabidopsis* through the floral dip method [51] and maize through the embryo infection method. T_0_- and T_1_-generation transgenic lines were screened with phosphinothricin (PPT) because the vector contains a screening marker gene *Bar*. The T_2_-generation positive transgenic plants were verified using PCR and a Bar protein test strip, and the expression levels of *ZmNAC89* in transgenic plants were confirmed through qRT–PCR.

All experiments were performed at the Molecular Breeding Experimental Base of Northeast Agricultural University located in Harbin City, Heilongjiang Province of China. The plants were stored in a specific room, and the locations where the plants were grown were burned once the experiments were concluded.

### 4.8. PCR and RT-PCR Analysis of Transgenic Plants

Genomic DNA was extracted from maize plants using the CTAB procedure. Due to the *ZmNAC89* gene being an endogenous gene in maize, primers were designed with the sequences of the promotor and the terminator of the plant expression vector. For PCR assays of *the ZmNAC89* gene, a 1758 bp fragment was amplified using the primers Pro-ZmNAC89-F/ZmNAC89-Ter-R (Appendix A). The *Bar* gene was assayed by using the primers of Bar transgenic plants.

TRIzol reagent (Tiangen, Beijing, China) was used to extract total RNA from 100 mg of maize seedling leaves or roots. The RT Reagent Kit (Transgene, Beijing, China) was used to perform reverse transcription on 500 ng DNase-treated RNA. Five-fold diluted cDNA was then used for PCR. A 430 bp *ZmNAC89* fragment was amplified with gene-specific primers from the promotor and the conserved region of the target gene. A 429 bp fragment was amplified using the *Bar* gene primer in the same manner described above. RT-PCR used the β-Actin gene as an internal control. Three replicates were performed.

### 4.9. Salt and Alkalinity Stress Treatment of Arabidopsis and Maize during Seedling Growth

For the salt and alkalinity tolerance assays of *Arabidopsis*, T_2_-generation transgenic seeds and the wild-type (WT) seeds were plated on MS agarose medium (0.8%). After 4 days of vernalization at 4 °C, plates were vertically cultured for 5 days in the incubator. The seedlings with consistent growth were selected and placed in an MS medium with various levels of NaHCO_3_ (0.5 and 1 mM) and NaCl (100 and 150 mM). After 14 days of vertical culture, the Epson Perfection V800 root scanner was used to analyze root parameters, including total root length (cm), root surface area (cm^2^) and root volume (cm^3^). Seven-day-old seedlings were placed in pots with a mix of vermiculite and nutrient soil in a greenhouse (22 °C, humidity 70%, 16 h light/8 h dark), grown for three weeks and irrigated at the same volume of 300 mM NaCl and 60 mM NaHCO_3_ solution every two days for two weeks.

For the salt and alkalinity tolerance assays of maize, T_2_-generation transgenic seeds and non-transgenic seeds were soaked for 6 h in 160 mmol/L NaCl solution, 25 mmol/L Na_2_CO_3_ solution and a saline–alkali mixed solution with a Na^+^ concentration of 100 mmol/L made up of NaCl, Na_2_SO_4_, NaHCO_3_ and Na_2_CO_3_ at a ratio of 1:9:9:1, and then germinated on filter paper in a culture dish. The germination potential was measured on the 4th day, and the fresh weight, root length, germination rate, dry weight and bud length were measured on the 7th day. T_2_-generation transgenic seeds, as well as non-transgenic seeds, were placed in 15 cm diameter pots with sand, with 5 seedlings in each pot, 30 seedlings per treatment and 3 replicates. The third-leaf-stage maize seedlings were irrigated with 160 mmol/L NaCl solution, 25 mmol/L Na_2_CO_3_ solution and a saline–alkali mixed solution with a Na^+^ concentration of 100 mmol/L made up of NaCl, Na_2_SO_4_, NaHCO_3_ and Na_2_CO_3_ at a ratio of 1:9:9:1, once every two days. All plants were irrigated with 1/2 Hoagland solution before treatment to prevent excessive salt accumulation in the sand.

### 4.10. Identifying Biochemical and Physiological Characteristics and Changes in Phenotype Associated with Alkalinity and Salt Tolerance

The relative conductivity, chlorophyll and proline content of *Arabidopsis* and maize *ZmNAC89*-overexpressed transgenic lines were then determined. The leaves of 0.05 g seedlings were immersed in 5 mL of deionized water. A conductivity meter (DDS-307) was used to detect the relative conductivity, which is the ratio of the electrical conductivity of exudates after soaking for 24 h at room temperature to the electrical conductivity of exudates after removal from a boiling water bath for 10 min until cooled.

Chlorophyll content was estimated according to the method used by Arnon et al. [52]. The plant leaf chlorophyll was extracted with 95% ethanol solution. The extraction absorbance was measured at wavelengths of 645 and 663 nm with a 754 UV-visible spectrophotometer (Shanghai Jinghua Company). The chlorophyll content was calculated according to the following formulas:$$\mathrm{Chlorophyll}\;a\;\mathrm{concentration}\;(\mathrm{mg}/L):\;C_{a}=12.7A_{663}-2.69A_{645}$$
$$\mathrm{Chlorophyll}\;b\;\mathrm{concentration}\;(\mathrm{mg}/L):\;C_{b}=22.9A_{645}-4.68A_{663}$$
$$\mathrm{Total}\;\mathrm{chlorophyll}\;\mathrm{concentration}\;(\mathrm{mg}/L):\;C_{a+b}=C_{a}+C_{b}$$

Proline content was measured according to the method of Bates et al. [53]. In brief, 5 mL of 3% sulfosalicylic acid was ground into 0.2 g plant leaves to extract proline. It was then placed in boiling water for 10 min, cooled, filtered and diluted to 5 mL. Then, 2 mL of acid ninhydrin and 2 mL of glacial acetic acid solution were added, followed by boiling for another 30 min. Next, 4 mL of toluene was added after it cooled, and the solution was gently vortexed. The toluene layer absorbance was measured at thebwavelength of 520 nm using a 754 UV-visible spectrophotometer (Shanghai Jinghua Company, Shanghai, China). The proline content was calculated from the standard curve.

### 4.11. RNA Extraction, Transcriptome Library Construction, and Sequencing

WT and transgenic maize seeds of the same size were selected, 75% were disinfected for 15 min and then they were washed with distilled water 3 times. Seeds were germinated via the sprouting paper method for 48 h, and seeds with the same budding potential were selected and cultured in a plastic bowl with a diameter of 15 cm until the three-leaf single stage. They were treated with 160 mmol/L NaCl solution and 25 mmol/L Na_2_CO_3_ solution, respectively. Before stress treatment, 1/2 Hoagland solution was irrigated once to prevent excessive accumulation of NaCl and Na_2_CO_3_ in the sand. After 3 h of stress, the aggregate tissues of 3 three-leaf maize seedlings were collected for RNA-seq analysis. Total RNA was isolated using TRIzol (Invitrogen, Carlsbad, CA, USA). The concentration and integrity of the RNA were evaluated using a NanoDrop ND-1000 (NanoDrop, Wilmington, DE, USA) and Bioanalyzer 2100 (Agilent, Santa Clara, CA, USA). The construction and sequencing of the transcriptome library were performed by Lianchuan Biotechnology Co., Ltd. in Guangzhou, China. The mRNA was enriched with magnetic beads connected with Oligo(dT) and then fragmented. Double-stranded cDNAs were synthesized using RNaseH, DNA Polymerase I and random hexamer primers. The cDNA fragments were purified using AMPure XP beads. The poly (A) addition was repaired and enriched via PCR to construct the cDNA library. The cDNA library was sequenced on the Illumina Novaseq™ 6000, and the reading length of sequencing was double-ended 2 × 150 bp (PE150).

Cutadapt (https://cutadapt.readthedocs.io/en/stable/ accessed on 19 March 2023) was used to filter out unqualified sequences and obtain valid data from the reference genome comparison. Statistical analysis was performed using HISAT2 (https://daehwankimlab.github.io/hisat2/ accessed on 19 March 2023). StringTie (http://ccb.jhu.edu/software/stringtie/ accessed on 19 March 2023) was used to initially assemble genes or transcripts, combine the initial assembly results of all samples and detect transcripts with GffCompare (http://ccb.jhu.edu/software/stringtie/gffcompare.shtml accessed on 19 March 2023) to obtain the final assembly annotation results. The Balltown package was used to provide a file input for fragments per kilobase of transcript per million mapped reads (FPKM) quantification.

The differentially expressed genes (DEGs) were identified with |log2foldchange| ≥ 1, *p* < 0.05. The DEG levels were normalized by calculating the fragments per kilobase of transcript per million mapped reads (FPKM). Gene Ontology (GO) analysis was performed for functional gene classification, while KEGG enrichment analysis was performed to understand the metabolic pathways involved in DEGs.

### 4.12. Sequence Variation Analysis of ZmNAC89

The genome-wide sequence of the maize inbred line B73 RefGen_v4 was checked in the MaizeGDB database (http://www.maizegdb.org/ accessed on 15 March 2023). Specific primers were designed for the candidate gene *ZmNAC89* using primer 5.0 software and amplified in 140 maize inbred lines. DNAMAN and DNA SPV6.0 software was used for the multi-sequence alignment of the *ZmNAC89* gene-coding region in different inbred lines, SNP nucleotide diversity analysis and haplotype diversity analysis. The primers are listed in Appendix A.

## 5. Conclusions

In this study, the *ZmNAC89* gene of the NAC family excavated by using RNA-seq was selected as the research object, and it was confirmed that it had transcriptional activation activity and was located in the nucleus. Under the four abiotic stress treatments, its expression was upregulated. Under salt–alkali treatment of maize and *Arabidopsis thaliana*, transgenic strains showed higher plant height and higher flowering, and showed good root characteristics. The differentially expressed genes identified by using RNA-seq in transgenic strains were identified in the REDOX pathway through GO enrichment analysis. Quantitative results indicated that *ZmNAC89* may respond to saline–alkali stress through the REDOX pathway or ABA signal transduction pathway. Among 140 maize inbred lines, HAP20 was found to be an excellent haplotype with salt and alkali tolerance, providing a material basis for saline–alkali-tolerance breeding. The results are helpful to further clarify the mechanism of saline–alkali tolerance in maize and to support marker-assisted selection in breeding.

## Figures and Tables

**Figure 1 ijms-24-15099-f001:**
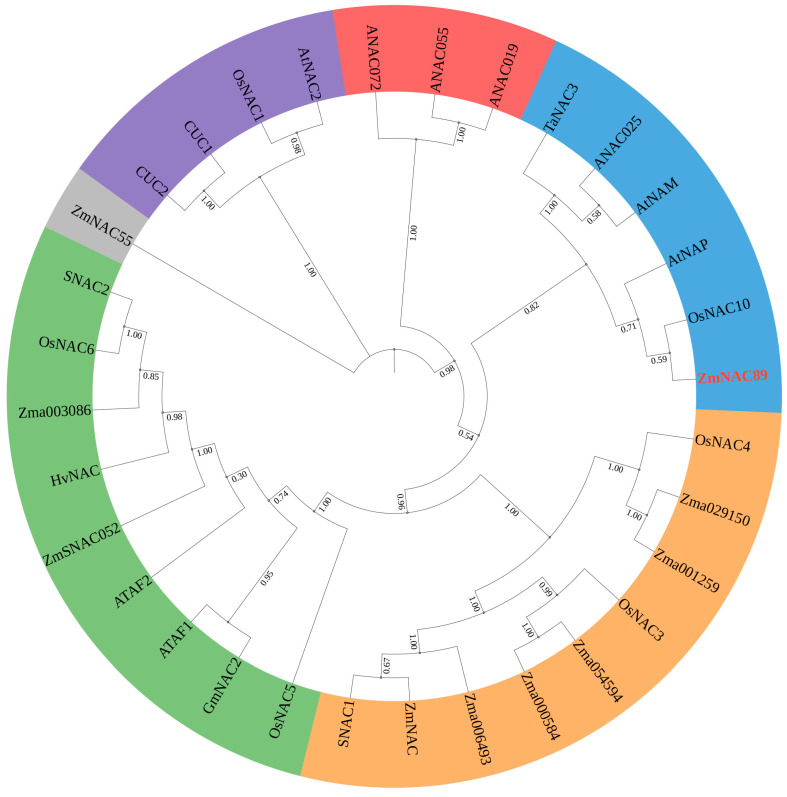
Phylogenetic tree analysis of the ZmNAC89 protein. The phylogenetic tree was constructed using MEGA (X) software based on 31 orthologs of ZmNAC89 in different species. The numbers on the tree nodes represent bootstraps from 1000 replicates.

**Figure 2 ijms-24-15099-f002:**
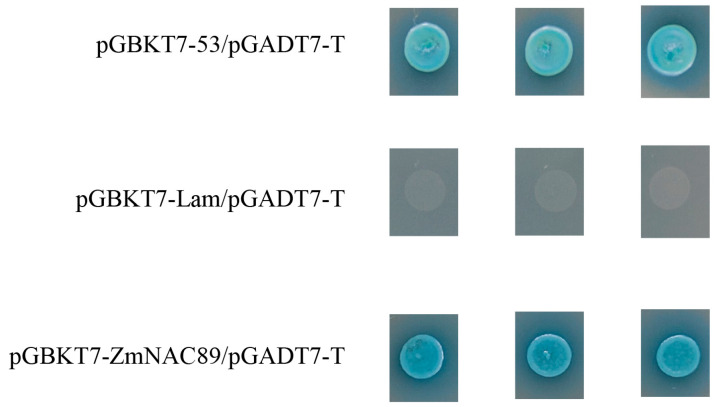
Transactivation activity analysis of *ZmNAC89* in yeast.

**Figure 3 ijms-24-15099-f003:**
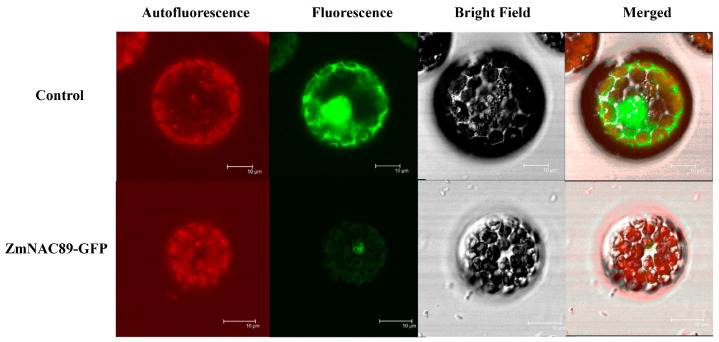
Subcellular localization of ZmNAC89-GFP protein in *Arabidopsis* protoplasts. Bars = 10 μm.

**Figure 4 ijms-24-15099-f004:**
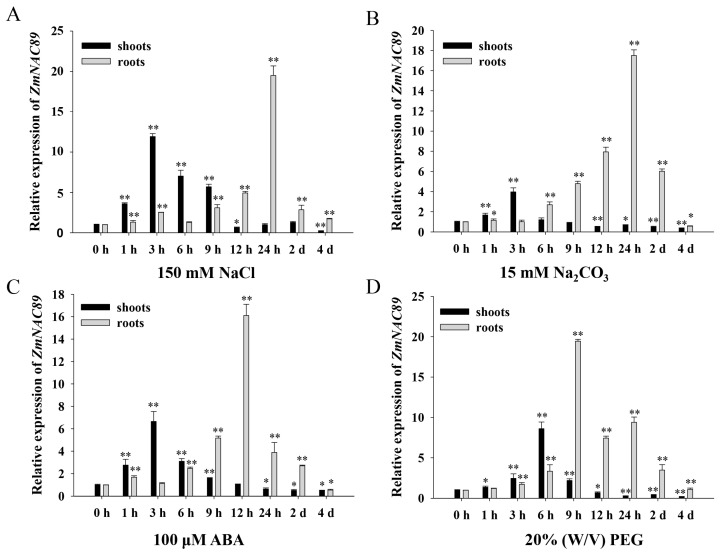
The relative expression of *ZmNAC89* at various time points under different stresses: NaCl stress (**A**), Na_2_CO_3_ stress (**B**), ABA stress (**C**) and PEG stress (**D**). *Actin* was used as a quantitative control. Bars indicate the standard error of the mean. * and ** indicate significance at *p* < 0.05 and *p* < 0.01, respectively.

**Figure 5 ijms-24-15099-f005:**
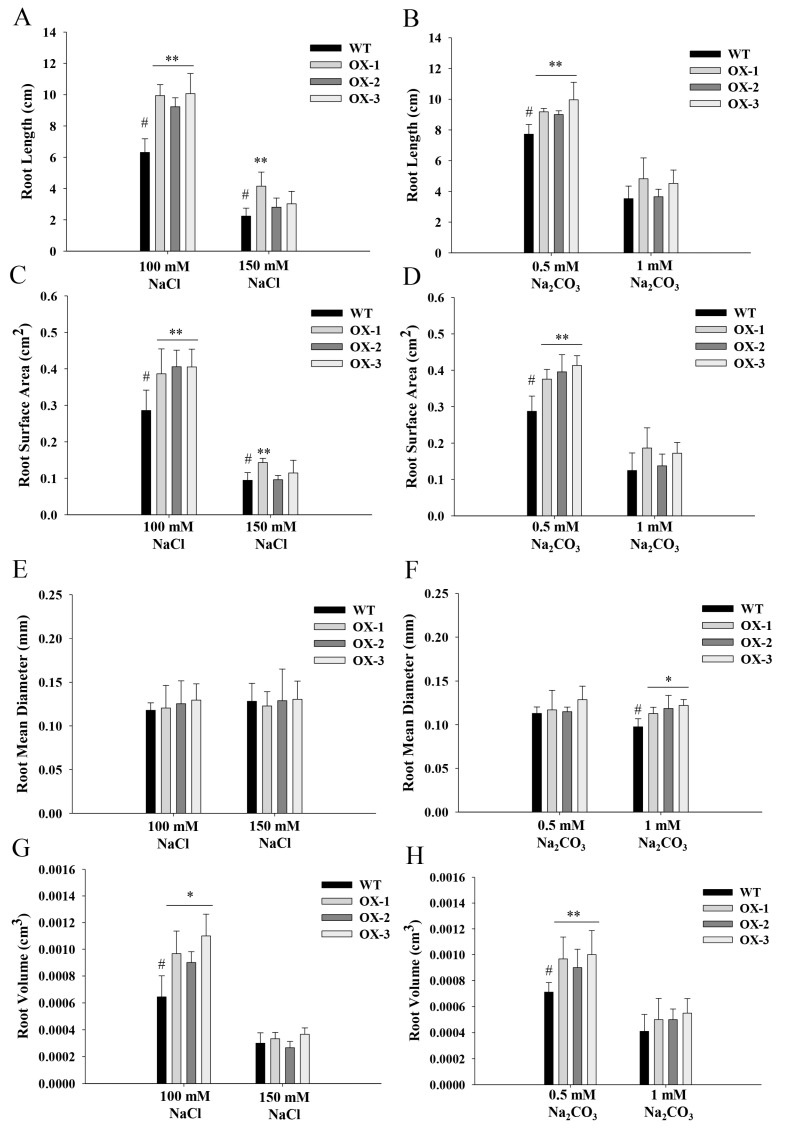
Overexpression of the *ZmNAC89* gene increased the resistance of *Arabidopsis*. (**A**,**B**) Root length of transgenic line with the *ZmNAC89* gene under two different stress treatments (unit: cm). (**C**,**D**) Root surface area of transgenic line with the *ZmNAC89* gene under two different stress treatments (unit: cm^2^). (**E**,**F**) Root mean diameter of transgenic line with the *ZmNAC89* gene under two different stress treatments (unit: mm). (**G**,**H**) Root volume of transgenic line with the *ZmNAC89* gene under two different stress treatments (unit: cm^3^). Bars indicate standard error of the mean. # is the comparison group of significance. * and ** indicate significance at *p* < 0.05 and *p* < 0.01, respectively.

**Figure 6 ijms-24-15099-f006:**
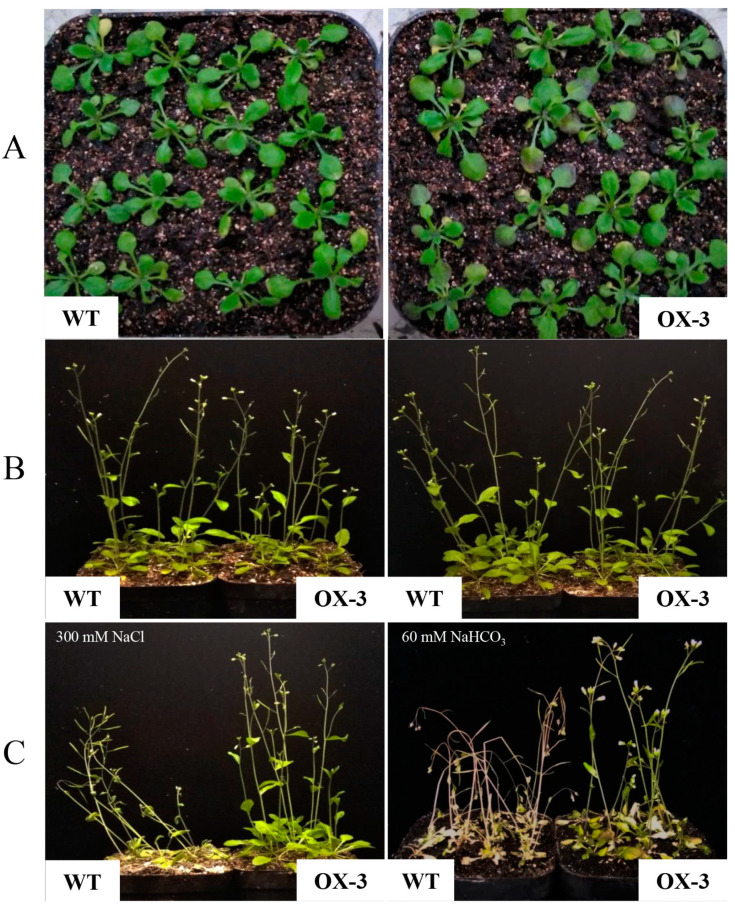
Phenotypic identification of Arabidopsis transgenic ZmNAC89 lines (OX-3). (**A**) Comparison diagram of germination stage, with receptor control on the left and transgenic strain on the right. (**B**) Comparison diagram of maturity; the left side is the receptor control, and the right side is the transgenic line. (**C**) Comparison diagram of stress treatment (300 mM NaCl and 60 mM NaHCO_3_); the left side is the receptor control, and the right side is the transgenic line.

**Figure 7 ijms-24-15099-f007:**
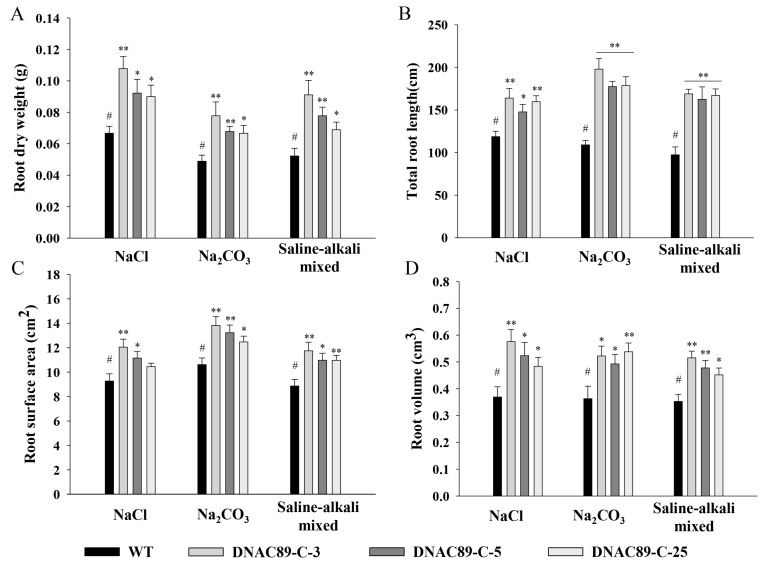
Overexpression of the *ZmNAC89* gene increased the resistance of maize. The T_2_-generation transgenic seeds and non-transgenic seeds were soaked for 6 h in 160 mmol/L NaCl solution, 25 mmol/L Na_2_CO_3_ solution and a saline–alkali mixed solution with Na^+^ concentration of 100 mmol/L combining NaCl, Na_2_SO_4_, NaHCO_3_ and Na_2_CO_3_ at a ratio of 1:9:9:1, and then germinated on filter paper in a culture dish. (**A**) Root dry weight of transgenic line with *ZmNAC89* gene under three different stress treatments (unit: g). (**B**) Total root length of transgenic line with *ZmNAC89* gene under three different stress treatments (unit: cm). (**C**) Root surface area of transgenic line with *ZmNAC89* gene under three different stress treatments (unit: cm^2^). (**D**) Root volume of transgenic line with *ZmNAC89* gene under three different stress treatments (unit: cm^3^). Bars indicate standard error of the mean. # is the comparison group of significance. * and ** indicate significance at *p* < 0.05 and *p* < 0.01, respectively.

**Figure 8 ijms-24-15099-f008:**
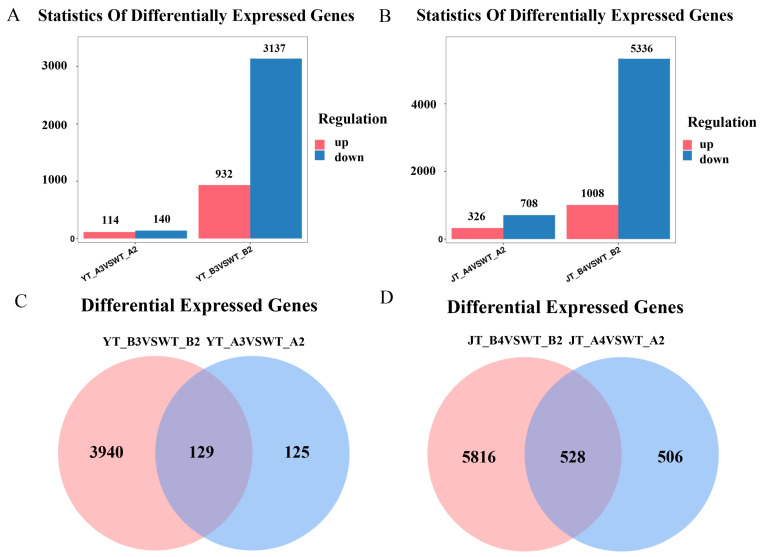
Transcriptomics data analysis of the transgenic *ZmNAC89* maize lines and WT under NaCl and Na_2_CO_3_ conditions. (**A**) Under NaCl conditions, the bar chart of differentially expressed genes in the receptor control comparison group (left) and the bar chart of differentially expressed genes in the transgenic comparison group (right). (**B**) Under Na_2_CO_3_ conditions, the bar chart of differentially expressed genes in the receptor control comparison group (left) and the bar chart of differentially expressed genes in the transgenic comparison group (right). (**C**) Venn diagram of differentially expressed genes in the transgenic comparison group and the recipient control comparison group under NaCl condition, with the middle overlap being the common genes in the two comparison groups (**D**) Venn diagram of differentially expressed genes in the transgenic comparison group and the recipient control comparison group under Na_2_CO_3_ condition, with the middle overlap being the common genes in the two comparison groups. edgeR software (3.42.4) was used to analyze the DEGs (|log2foldchange| ≥ 1, *p* < 0.05).

**Figure 9 ijms-24-15099-f009:**
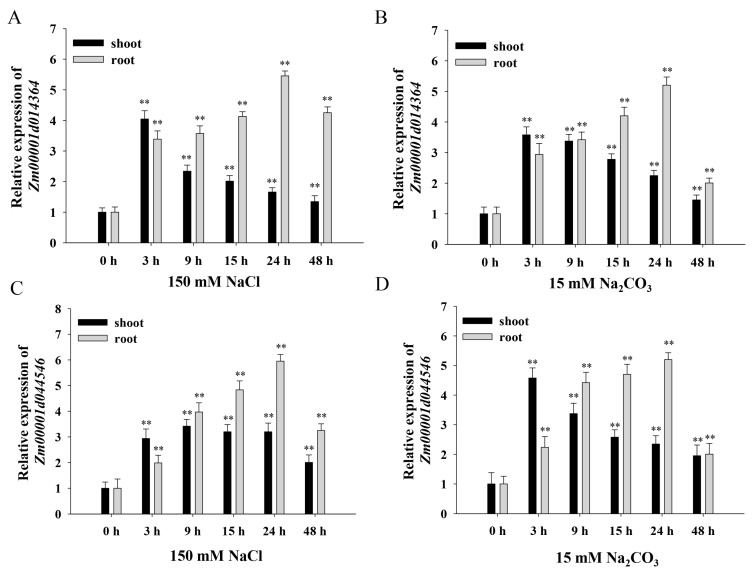
The relative expression of *Zm00001d014364* and *Zm00001d044546* in leaves and roots of transgenic lines under stress treatment at different times. (**A**): Relative expression of *Zm00001d014364* in different parts treated with 150 mM NaCl. (**B**): Relative expression of *Zm00001d014364* in different parts treated with 15 mM Na_2_CO_3_. (**C**): Relative expression of *Zm00001d044546* in different parts treated with 150 mM NaCl. (**D**): Relative expression of *Zm00001d044546* in different parts treated with 15 mM Na_2_CO_3_. *Actin* was used as a quantitative control. Bars indicate the standard error of the mean. ** indicates significance at *p* < 0.01.

**Figure 10 ijms-24-15099-f010:**
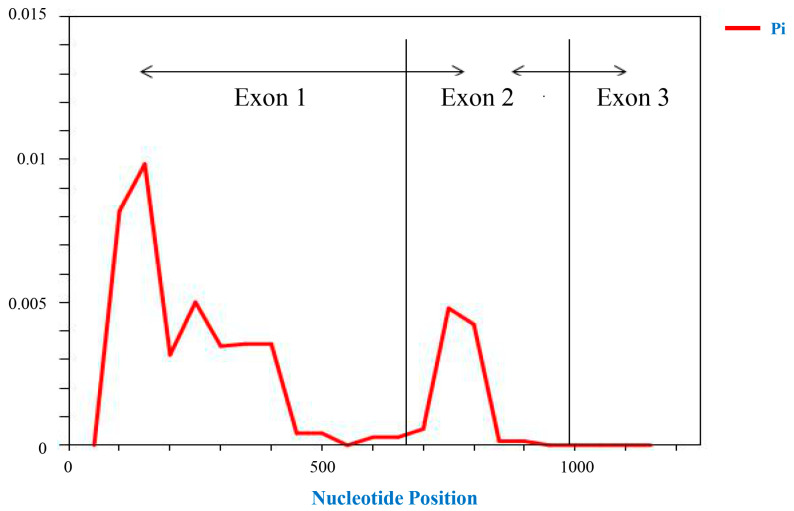
Diversity of nucleotides in the *ZmNAC89* gene coding region.

**Table 1 ijms-24-15099-t001:** Determination of physiological parameters of T_2_ transgenic lines with *ZmNAC89* gene under 300 mM NaCl stress.

Line	Relative Conductivity (%)	Chlorophyll (mg·g^−1^)	Proline (μg·g^−1^)
WT	22.52 ± 2.11	7.45 ± 0.59	14.29 ± 0.21
OX-1	15.87 ± 3.07 **	10.18 ± 0.41 **	20.28 ± 0.35
OX-2	15.82 ± 1.52 **	10.44 ± 0.45 *	23.85 ± 1.91 *
OX-3	13.78 ± 2.56 **	11.08 ± 0.43 **	29.09 ± 0.69 **
OX-4	14.99 ± 1.48 **	11.46 ± 0.65 **	28.22 ± 0.71 **
OX-5	20.04 ± 1.66 *	10.87 ± 0.50 **	23.13 ± 1.32 *
OX-6	17.31 ± 0.41 **	10.63 ± 0.45 **	21.39 ± 0.45

Values are mean ± SE (*n* = 3); * and ** indicate significance at *p* < 0.05 and *p* < 0.01, respectively.

**Table 2 ijms-24-15099-t002:** Determination of physiological parameters of T_2_ transgenic lines with *ZmNAC89* gene under 60 mM NaHCO_3_ stress.

Line	Relative Conductivity (%)	Chlorophyll (mg·g^−1^)	Proline (μg·g^−1^)
WT	35.05 ± 0.07	5.48 ± 0.14	10.93 ± 0.91
OX-1	20.70 ± 1.41 **	7.75 ± 0.23 **	18.18 ± 0.35 *
OX-2	19.14 ± 1.45 **	8.54 ± 0.95 **	21.69 ± 1.25 **
OX-3	18.74 ± 2.30 **	9.12 ± 0.75 **	24.96 ± 0.31 **
OX-4	18.80 ± 0.44 **	9.30 ± 0.22 **	24.55 ± 0.34 **
OX-5	26.50 ± 2.34 **	7.95 ± 0.26 **	20.65 ± 0.59 **
OX-6	21.31 ± 3.08 **	7.72 ± 0.36 *	18.13 ± 0.22 *

Values are mean ± SE (*n* = 3); * and ** indicate significance at *p* < 0.05 and *p* < 0.01, respectively.

**Table 3 ijms-24-15099-t003:** Analysis of nucleotide diversity.

Material Category	Number of Materials and Tolerance	Nucleotide Diversity (pi)	Haplotype Diversity (HD)	Tajima’s D	Fu and Li’s D *	Fu and Li’s F *
Na_2_CO_3_	51 medium-resistance materials	0.00291	0.846	−1.10181	−1.83401	−1.87261
	33 high-resistance materials	0.00205	0.753	−0.06472	−0.76798	−0.64457
	23 resistant materials	0.00286	0.819	−0.52477	−1.02437	−1.01949
	17 sensitive materials	0.00127	0.742	−0.53609	−0.7063	−0.75738
	16 highly sensitive materials	0.00266	0.842	−0.15008	−0.5959	−0.54408
NaCl	45 resistant materials	0.0024	0.798	0.01116	0.51653	0.41226
	30 sensitive materials	0.00216	0.828	−1.03859	−2.03901	−2.02296
	27 highly sensitive materials	0.00307	0.82	−0.82036	−0.32925	−0.56118
	20 medium-resistance materials	0.00245	0.818	−0.38437	−1.50627	−1.36738
	18 high-resistance materials	0.00222	0.846	−0.67509	−0.64189	−0.75081

* indicate significance at *p* < 0.05 respectively.

## Data Availability

The transcriptome sequencing data have been deposited in the Sequence Read Archive (SRA) at the National Center for Biotechnology Information (NCBI) under the accession number PRJNA1026668.

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
