# Peer review of "The Transcription Factor ZmNAC89 Gene Is Involved in Salt Tolerance in Maize (Zea mays L.)"

_ijms, 2023, doi:10.3390/ijms242015099_

Round 1

Reviewer 1 Report

Comments In general, the submitted manuscript is set out to explore the role of ZmNAC89 gene in salt tolerance in maize (Zea mays L.). I trust that the paper has the publication potential, however it should be improved in various aspects, as mentioned in the following comments:

Why authors choose to study NAC gene family member? In my opinion it has already been extensively explored for stress adaptation in plants.

Why ZmNAC89 expression was upregulated under ABA treatment?

Which were the genes that are involved in oxidation-reduction process in this study?

Always (where applicable) use the significant letters to highlight the significance.

Please mention the osmotic potential for PEG treatment.

Discussion section is too small and conclusion is also not included in this study.

Recomendation: Major revision

 Minor editing of English language required

Reviewer 2 Report

The authors in their manuscript entitled “A transcription factor of the ZmNAC89 gene is involved in salt 2 tolerance in maize (Zea mays L.).” present results regarding genomic, transcriptomic, gene expression and functional characterization of the ZmNAC89 transcription factor.

The aim of research is clear, and the methodological approach and implementation are correct. The discussion of the results and subsequent conclusions support the main findings.

Major revision is suggested for the following parts:

1.      Please correct the manuscript title since ZmNAC89 is “the” transcription factor. As written in the present form the reader understands that the authors are going to talk about “a” transcription factor that regulates the ZmNAC89 gene (which is also a TF)

2.      Please (must) reform and enlarge figures 3, 4, 5, 7, 8 and 9 so that they will be legible in an A4 printed version of the manuscript. Even enlarged in a PC screen they cannot be read/interpreted easily. Please do various printing tests to confirm this is achieved.

3.      In each of the figure legends please incorporate more details regarding particularly the salt concentrations. The reader should be able to interpret the figure reading the legend and not have to go forward to MM and the back to the figure to make the connection. All relevant information should be in the legend so that every figure should be interpreted as a stand-alone concept. Please improve descriptions in axes of each figure and put a title (if possible) above each individual (e.g. 7A, 7B, etc.) figure.

4.      In figure 6 in A, B and C images please indicate what plant(s) are in each pot inside each individual picture.

5. Please describe in more detail the data results regarding DEG data (lines 233 to 240), since this section is relatively poor. A more detailed description would result in a more luminous version of the manuscript

Minor editing of English language required

Reviewer 3 Report

The work by Hu et al. advances our understating of salt tolerance and its genetic, regulatory and functional bases in maize. The manuscript demonstrates the involvement of a transcription factor from the ZmNAC89 gene in the salt tolerance response in maize. The report is well written and condensed, as well as technically appropriate. However, before being able to recommend acceptance, I invite authors to address the following amendments.

First, the introduction section should properly close (in L68 before the study goal) with an explicit research questions, hypotheses and expected results (in L73). Pleas emind that so far only a main goal is described in L69. For instance, do authors hypothesize a polygenic or a Mendelian/biomodal segregation for salt tolerance in maize? Are traits correlated with salt tolerance expected to exhibit antagonistic pleiotropy or conditional neutrality as the result of the regulation derived from the ZmNAC89 transcription factor?  

Second, methodological speaking authors should comment more clearly how they handled temporal and environmental variation on the transcriptomics within the study. After all, quantitative trait segregation in Salt tolerance (something authors must shown in a fir main histogram) can be accounted both by the polygenic model and by reaction norms triggered by environmental gradients.

Third, concerning the results, figures and tables are very well edited and insightful. Still, authors should improve the picture resolution and quality in figures 2, 3 and 6, as well as the size of the letters and numbers in the axes of the histograms in figures 4, 5, 7 and 9. This readability is key for interpretation.

Fourth, results must also include a second section in L303 briefly commenting on a posteriori statistical power analysis, specially taking into account that the strong influence of temporal and environmental sampling on transcriptomic and phenotypic variance. How representative are they? Furthermore, when salt stress effects are measured and observed is a key issue to consider. Was the salt tolerance observed during the flowering phenological phase? What about terminal stress? Please expand, especially for readers coming more from a breeding side. 

Fifth, please explicitly re-visit the above sampling caveat within the discussion section (e.g., L341). Additionally, the discussion, although perceptive, should embrace broader reflections on whether salt tolerance in maize could be understood as a plastic (via trans generational epigenetic inherence mediated by the ZmNAC89 transcription factor) or alternative a genetic adaptive strategy to cope with osmotic imbalance. Besides, since abiotic stresses are usually pleiotropic, please also comment in the discussion (as follow up to the insights in L46 results, and the results reported in section 2.1 at L116) on potential gene functional correlates of salt tolerance with drought, heat and flooding tolerance (consider referring to Front Genet 2019 10:954), which are stresses typically correlated with salt stress in the face of hypoxia and osmotic imbalance (look into and consider referring to PLoS One 2013 8(5):e62898). Please comment/discuss the latter list of correlations in the abiotic responses with salt tolerance in the light of ABA-mediated drought tolerance pathways such as ASR genes (consider referring to BMC Genet 2012 13:58, as a follow up of the results presented in L118), ERECTA-mediated drought tolerance (i.e., consider referring to Plant Sci 2016 242: 250), and DREB-based ABA-independent drought tolerance responses (consider referring in L333 to the work Theor Applied Genet 2012 125(5):1069-85). Any further insights in this regard in the studied maize background?

As complement to my previous comment, what about combined salt + drought + heat treatments to study the maize background with known salt tolerance properties? Also related to this point on pleiotropy, authors must also briefly mention whether adaptive trade-offs for salt tolerance are observed/predicted (check and comment in the light of e.g. Front Plant Sci 2018 9:128). For instance, a biomass/reproductive trade-off triggered by salt stress in maize may also be evident in more subtle ways.

Sixth, although the paper provides good evidence into the genetic regulatory mechanisms for salt tolerance in maize, a major question that authors should prospect in their discussion is how maize improvement for salt tolerance (as a follow up to L54) may unlock and effectively utilize the identified trait variation (see as guidance and consider referring to Genes 2021 12:556). Gene editing, recurrent backcrossing and inter-specific schemes (condensed, contrasted and discussed in e.g. Front Genet 2020 11:564515, and Genes 2021 12:783, both recommended) may offer an avenue in this regard that authors should acknowledge (consider referring to Agronomy 2021 11:1978). Please envision any other recommendation (some ideas in the next paragraph).

Last but not least, the report is so far lacking a very brief closing paragraph (L341) that describes the major caveats/limitations of the present work. It is never beyond the scope of any research to explicitly acknowledge the study caveats, especially when dealing with highly variable trait profiles in a complex trait with strong GxE effects and plasticity such as salt tolerance. For instance, trait variation may exhibit antagonistic pleiotropy or conditional neutrality when expression profiles in contrasting environmental treatments, something to be tested in more profound trials). Also link to this point, a short perspectives section, just after the discussion in L341, would be insightful for readers to fill the identified caveats as part of future research. Specifically, what other maize backgrounds, traits and combined abiotic stresses must future studies target? How could the identified ZmNAC89 regulatory transcript factor source maize pre-breeding and future improvement programs?

Finally, include an explicit conclusions section after the new perspectives section in L341.  

Round 2

Reviewer 2 Report

I thank the authors for their revised manuscript.

Still, figures 5, 7, 8 and 9 are illegible. Please revise the set-up, probably better in a vertical format with larger individual images and figures. Letters and number figures are illegible. To confirm, please try to print in A4 and read the figures.

Figure 6 cannot be interpreted since it is not clear what the authors mean by “left” and “right” side. As mentioned in the first round of comments “inside each individual picture please indicate on each pot what plant is.”.

It is important to clarify this issue because results cannot be interpreted, consequently it is not clear what is discussed, and this may lead to wrong conclusions.

Figure 6 cannot be interpreted since it is not clear what the authors mean by “left” and “right” side. As mentioned in the first round of comments “please indicate on each pot what plant is”.

It is important to clarify this issue because results cannot be interpreted, consequently it is not clear what it is discussed, and this may lead to wrong conclusions.

Reviewer 3 Report

The improvements have been substantial. I thanks the authors for this thoughtful revision. From my side it would have green light 

Author Response

Thank you very much for the excellent and professional revision of our manuscript.

Round 3

Reviewer 2 Report

I have no further queries or comments. Thank you.

Minor editing of English language required